# Influence of the Additive of Ceramic and Intermetallic Powders on the Friction Properties and Temperature of the Wet Clutch Disc

**DOI:** 10.3390/ma15155384

**Published:** 2022-08-04

**Authors:** Aleksander Yevtushenko, Michal Kuciej, Piotr Grzes, Aleksander Ilyushchanka, Andrey Liashok

**Affiliations:** 1Department of Mechanics and Applied Computer Science, Faculty of Mechanical Engineering, Bialystok University of Technology (BUT), 45C Wiejska Street, 15-351 Bialystok, Poland; 2The State Scientific Institution “Powder Metallurgy Institute” (SSI PMI) of the National Academy of Sciences of Belarus, 41 Platonova Street, 220005 Minsk, Belarus

**Keywords:** friction material, clutch, temperature, frictional heating, finite element analysis

## Abstract

The basic function of friction clutches is to transfer the torque in the conditions of its smooth engagement without vibrations. Hard working conditions under high thermal and mechanical loads, leading to high temperature in the contact area, intense wear, and instability of the coefficient of friction impose restrictive criteria in the design of friction materials. In this paper, the results of experimental research of the effect of ceramic and intermetallic additives to the copper-based material of the friction disc of the clutch on the thermophysical and frictional properties were presented. Next, these properties were incorporated in the proposed contact 3D numerical model of the clutch to carry out computer simulations of the heating process and subsequent cooling. Based on the obtained experimental data and transient temperature changes of the friction and steel discs, the relations between the powder additives, thermophysical properties of the five friction materials, and coefficients of friction, wear, and temperature reached were discussed. Among these, it was found that when working with the lubrication, the largest values of the coefficient of friction 0.068 and wear $13.5\;\;\mu m\;\;{\mathrm{km}}^{-1}$ were reached when using the 3 wt.% SiC additive.

## 1. Introduction

Powder sintered friction materials (PSFM) have become widely used in the friction units of the automotive vehicles, motorcycles, tractors, airplanes, boats, machine tools, etc. Such friction units include, in particular, hydro mechanical gearboxes, oil-cooled brakes, clutches, etc. [1,2]. Powder metallurgy allows obtaining composite materials using powders of different types and at various chemical compositions.

The PSFM should comply with the following requirements: stable value of the coefficient of friction, high wear resistance, effective adaptation, and high thermal conductivity [3]. As a rule, PSFM on the basis of copper are used to operate under lubrication conditions, while materials based on iron are used at dry friction.

The achievement of the given level of tribotechnical and operating properties of the PSFM has been achieved by the use of additives of various kinds, and granulometric composition, the content of which is within the range 0.5–15 vol.%. The additives are able to interact with the metal base, and to localize in the form of individual inclusions. The main additive used in the composition of PSFM are graphites of various types, as well as carbon-containing additives with an amorphous structure, such as coke and anthracite [4,5].

It has been shown that graphite with a size of 80 μm provides a high value of the coefficient of friction, compared with graphite with a size of 8–10 μm. However, a much greater increase in the coefficient of friction was achieved with the use of coke powder [6]. An additive in the form of graphite allows increasing the operating properties of the friction unit material [7,8,9]. For a material containing 5 masses % of graphite, an increase in the sliding velocity leads to a sharp increase in the coefficient of friction, while the material itself is characterized by the increased antifriction and anti-pressure properties, and performance under increased load-velocity conditions. For the tin bronze containing 10% of graphite, the wear resistance is significantly higher than bronze with the graphite contents [10].

However, the use of graphite by itself does not provide specific tribological properties. Their further improvement is achieved through the use of additives in the PSFM structure of solid ceramic powders and their compounds.

The composition of friction material (FM) for clutches and brake units, which has a high coefficient of friction and a small difference between dynamic and static coefficients of friction, was presented in the patent [11]. It was noted that this effect is achieved using 8–15% of ${\mathrm{Al}}_{2}O_{3}$. By means of a method of powder metallurgy, a metal-based FM class with high coefficient of friction and wear resistance, as well as reduced acoustic characteristics was created [12,13,14,15,16,17]. These materials contain in their composition 2–30% of the component of solid particles selected from metal oxides (composites), metal nitrides (carbonitrides), metal carbides, metal borides, intermetalides, and minerals with a Mohs hardness of 3.5 or more.

Titanium dioxide is widely used in industry as an addition of a small cost, characterized by stable properties and non-toxicity. This additive is characterized by a very high specific surface area, up to 600 $m^{2}\;\;g^{-1}$, and a low thermal conductivity [18].

The carbides of transition metals are characterized by high hardness and, at the same time, brittleness. The most commonly used is silicon carbide, which has high hardness and thermal conductivity at low density. Its main disadvantage is the low ($2-3\;\mathrm{MPa}\;\;m^{1/2}$) viscosity of destruction [19].

Currently, the use of the intermetallic powder additives in tribotechnical materials is of great interest. The intermetalides ${\mathrm{Ni}}_{3}\mathrm{Al}$ and NiAl appear in difficult operating conditions due to the set of unique properties, such as increased value of impact toughness, resistance to oxidation at elevated temperatures and thermal resistance. The above-mentioned intermetalides have a density 7.3 and 5.9 $g\;\;{\mathrm{cm}}^{-3}$, respectively, are characterized by a high Young’s elasticity, and may be used in products for constructional and tribotechnical purposes [20]. An effect of the addition of intermetallic powder Ti-46Al-8Cr, obtained by the method of mechanoactivable self-propagating high-temperature synthesis, on the tribological properties of the copper-based antifriction material was investigated [21]. It was shown that an increase in the content of aluminite from 0.5% to 1% leads to a decrease in the intensity of wear of the material by more than 3 times. The inclusion of the additive of ${\mathrm{NiAl}/\mathrm{Al}}_{2}O_{3}$ powder system in the frictional material based on copper in the range of 0.5–2.5% revealed an increase in the dynamic coefficient of friction from 0.040 to 0.051, while wear ranged from 4.2 to 5.7 $\mu m\;\;{\mathrm{km}}^{-1}$ [22].

Newly designed PSFM materials used for friction elements of clutches, before implementation to production process, undergo a series of restrictive tests, both in full scale and laboratory tests. Even at the stage of their preselection and elimination of the least promising ones, this may be an expensive and time-consuming process. Numerical models are effective (time, costs) in this first phase of designing a new friction pair, allowing for a preliminary analysis of the level and temperature distribution of the friction pair. The novelty presented in this article is the proposed 3D numerical models to analyze the transient temperature fields of the new designed PSFM friction materials. The study presented an extensive numerical finite element (FE) model (from ref. [6]) of friction heating for the estimation of temperature distributions in a wet clutch. Unlike the numerical model from the article [6], this model takes into account the change of thermal properties of the steel disc under the influence of temperature and two different phases of the clutch operation, i.e., after its engagement (heat generation) and disconnection (cooling). The analysis of temperature distribution, based on the structural and tribological tests of newly developed PSFM materials, allowed selecting the most effective friction pair for the use in the wet clutch disc.

## 2. Materials and Experimental Methods

As a basis for the friction material, a mixture of copper powders (81 wt.%), tin (11 wt.%), and elemental graphite GE-1 (8 wt.%) was used. The initial mixture was prepared in a blade mixer by mixing, within 45 min, the copper powders obtained by electrolysis with a mean particle size of 100 μm (Figure 1a), the tin obtained by spraying the melt, with a mean particle size of 20 μm (Figure 1b), the elemental graphite GE-1, the natural origin obtained by extraction, grinding, and processing in acid solution and having a scaly shape with a mean size of 100 μm (Figure 1c). As test additives, the powders of silicon carbide with a size 4–9 μm (Figure 1d) and the titanium dioxide, which is a conglomerate with a size of 100–150 μm, were used. The conglomerate consisted of the ultra-disperse powders of predominantly spherical shape up to 0.2 μm in size (Figure 1e) and the intermetallic powder Ti-46Al-8Cr. Particles of the powder Ti-46Al-8Cr in the size of 50–500 nm formed agglomerates in the size 5–20 μm with a high specific surface area (Figure 1f). The micro hardness of the powder particles was 4000–5140 MPa. The powders were supplied by manufacturers.

The TI-46AL-8CR system was obtained by the method of the mechanoactivated self-propagating high-temperature synthesis (MASHS) [23]. The preliminary mechanical processing of the reaction mixture of the powders of the titanium, aluminum, and chromium was carried out in a mill A-4.5 with the following parameters: rotational speed of the impeller shaft $360\;\;r\;\;\min^{-1}$, the ratio of the mass of spheres and powder 10:1, and the duration of processing 3 h. The subsequent self-propagating high-temperature synthesis was carried out in the experimental reactor for MASHS in argon environment. The mixture of powders was ignited with a tungsten spiral heated by the passage of an electric current. After cooling, the resulting sinter was milled in the planetary mill Pulverisette 6 (Fritsch, Germany) in an alcohol medium at the following parameters: the diameter of the spheres 5 mm, the mass ratio of the spheres and the powder 20:1, rotational speed of the impeller drive shaft $400\;\;r\;\;\min^{-1}$, and the grinding time 30 min.

The synthesized material Ti-46Al-8Cr, according to X-ray diffraction, consisted of the basis in the form of intermetalide γ−TiAl (the spatial group P4/mmm) (Figure 2a,d phase 1), doped by chromium, containing 64–68 at. % Ti, 30–34 at. % Al and up to 5 at. % Cr, the inclusions of double intermetalides ${\mathrm{Ti}}_{3}\mathrm{Al}$ and ${\mathrm{AlCr}}_{2}$, and the triple intermetalide ${\mathrm{Al}}_{0.67}{\mathrm{Cr}}_{0.08}{\mathrm{Ti}}_{0.25}$ (Figure 2a,c). Thin secondary τ-phase ${\mathrm{Al}}_{0.67}{\mathrm{Cr}}_{0.08}{\mathrm{Ti}}_{0.25}$ smaller than 0.5 μm (Figure 2b, phase 3), falling out in grains of titanium monoaluminide, contained about 68–71 at. % Al, 20–25 at. % Ti, 7–12 at. % Cr and had a cubic grate of Pm-3m type, which provided coherence of boundaries with the *γ*-phase.

The $\alpha_{2}-{\mathrm{Ti}}_{3}\mathrm{Al}$ phase (the spatial group P63/mmc) was localized mainly along the boundaries of grains and contained about 2 at. % Cr (Figure 2b, the phase 2). In addition, at the grain boundaries of titanium monoaluminide, there were also inclusions of excessive phases of chromium compounds with aluminum and titanium containing 46–53 at. % Al, 45–60 at. % Cr, and 3–5 at. % Ti (Figure 2b, phase 4), the formation of which was probably due to the problems of diffusion redistribution of components in the SHS process under conditions of predominantly solid-phase interaction.

The samples of the friction discs for testing were made as follows: obtained charge from the initial powders was applied by free filling to the surface of the steel base using special technological equipment, and then preliminary sintering was carried out in dissociated ammonia at a temperature of 840 °C within 50 min. For forming a system of oil-removing channels and grooves on the surface of the sintered material, as well as obtaining a porosity of 12–18%, the sintered workpiece of the friction disc was subjected to plastic deformation (embossing) with a punch having a profile in the form of a “grid” on the surface. Then, the final sintering was carried out at a pressure of 0.1 MPa in a medium of dissociated ammonia, which contains 75% $H_{2}$ and 25% $N_{2}$ at a temperature of 840 °C within 3 h. The friction and steel discs are shown in Figure 3.

The study of the tribotechnical properties of the friction material was carried out on a friction machine IM-58 according to the scheme friction disc-counter body at the following input parameters. The initial velocity of braking was $10\;\;m\;\;s^{-1}$, the contact pressure was 4 MPa, the moment of inertia of rotating masses was $0.56\;\;N\;\;m\;\;s^{2}$, and the work of friction was 27.5 kJ. As a counterpart, a disc made of 65H steel with a hardness of 260–320 HB and a surface roughness of Ra = 0.7–0.8 were used [6]. The bedding-in (burnishing) of the working surfaces by 300 engaging cycles was carried out. Then, 10 measurements of the values of the coefficients of friction and wear were made. From these data, mean values were determined.

The investigation of structure was carried out by means of the optical microscope MEF-3 (Austria). The morphology of the surface of the friction disc and its microstructure were studied on a high-resolution scanning electron microscope MIRA (Czech Republic) with a micro-X-ray spectral console INCA 350 of the Oxford Instruments (UK) company. The phase composition was examined on an X-ray diffractometer Ultima IV (Rigaky) in Cu Kα-radiation at an X-ray tube voltage of 40 kV and the anode current of 40 mA. The parameters of the crystal grate of the alloys were determined by diffraction lines located at the large scattering angles. For a phase analysis, a standard PDF card files was used. The thermophysical properties of investigated compositions of friction materials were carried out on the analyzer of thermal properties Hot Disk TPS2500S. As a sensor, a spiral, being a source of heat, was used. The sensor was located between the sample under the study and the sample, with the known thermophysical properties. Ten measurements were made after a given period of time, and the mean values of the thermophysical properties were established. The tested samples had a diameter of 50 mm, a thickness of 10 mm, and were obtained by compressing at the pressure of $2.5\;\;t\;{\mathrm{cm}}^{-2}$ and sintering at 840 °C for 3 h.

## 3. Results of Experimental Investigations

The results of the study of the physical and frictional properties of five compositions of friction materials are given in Table 1.

The data obtained showed that use of additive SiC obtains the greatest value of the coefficient of friction. The solid inclusions of SiC in the process of friction are crumbled, displacing coarsely dispersed graphite from the surface of the friction material. The change in the morphology of the surface layer, the closure of pores, and increase in the area of the metal phase were fixed (Figure 4b).

The introduction of the additive of the intermetallic powder of the Ti-46Al-8Cr in an amount of 2 wt.% showed an increase in the coefficient of friction to 0.055, whereas for the basic composition, without additives of powders, it was 0.036. An analysis of the morphology of the surface layer showed that the initial porosity of the friction material was preserved, and there is no replacement of graphite particles (Figure 4c), which is characteristic of the basic composition of the friction material (Figure 4a).

The use of TiO_2_ powder additive in an amount of 2 wt.% and 5 wt.% led to an increase in the coefficient of friction to 0.043 and 0.052, respectively. An increase in the addition of TiO_2_ powder from 2 to 5 wt.% showed a change in the morphology of the friction surface of the friction material with a slight increase in the area of the metal phase (Figure 4d,e).

## 4. Numerical Simulation of the Temperature Mode of the Clutch

### Operating Parameters

The aim of the numerical simulations was to investigate an effect of the above-mentioned powder additives, namely one ceramic (SiC) denoted as variant 1 and three intermetalides (2–Ti-46Al-8Cr, 3–2 wt.% TiO_2_ and 4–5 wt.% TiO_2_) to 0—the friction base material, on the clutch temperature, presented in Figure 3. The analyzed friction pair consisted of two discs—a fixed one with a steel substrate (65H) and a friction material applied to it—and a steel (65H) disc rotating against the specimen. The thermophysical properties of the materials at the ambient (initial) temperature $T_{0}=20\;\;{}^{\circ}C$ are presented in Table 1. The changes in the properties of 65H steel with temperature increasing from 20 °C to 800 °C are shown in Table 2.

The dimensions of the clutch components and the initial kinetic energy of the system were the same as in the article [6] (Table 2). The calculations were carried out for five variants of friction materials: 0—basic, 1—TiC, 2—Ti-46Al-8Cr, 3—2 wt.% TiO_2_, 4—5 wt.% TiO_2_ with the corresponding values of the coefficients of friction listed in Table 1.

## 5. Heating Taking into Account the Thermal Sensitivity of 65H Steel (First Calculation Model)

Two 3D numerical models were developed using the finite element method (FEM) adapted in the Heat Transfer Module of the COMSOL Multiphysics^®^ programme. The first model was a generalization of the linear (with material properties unchanged) model from the article [6] for the case of thermally sensitive materials (with temperature-varying properties of 65H steel). The finite element analysis was limited only to the friction heating stage during braking. The results of the calculations are presented in Figure 5 and in Table 3.

The evolutions of temperature of the friction surfaces at the equivalent radius $r_{eq}=39.4\;\;\mathrm{mm}$ shown in Figure 5 for thermosensitive (dashed lines) and constant (solid lines) properties of materials revealed typical changes for braking at constant deceleration. Namely, temperature increased rapidly at the beginning, reached maximum value, and decreased until the stop. The obtained maximum temperature values did not exceed 165 °C (higher values appeared when taking into account thermosensitivity of the steel), hence the omission of the thermal sensitivity of the friction materials hardly influencing the simulation results. It should also be noted that in this temperature range (from 20 °C to 165 °C) the changes in the properties of steel 65H are negligible (Table 2).

Comparison of the results for 5 materials analyzed shows how braking time affects the maximum value of the temperature. Since frictional sliding lasts only a few seconds, generated heat cannot be absorbed by the components of the clutch and convection. Therefore, differences in braking times have a strong effect on the maximum temperature reached. The highest value (at thermosensitive material) is equal to 161.8 °C, whereas the lowest is equal to 123.1 °C.

## 6. Heating with Subsequent Cooling of the Clutch Elements (Second Calculation Model)

The second computational model concerned both the clutch heating stage due to friction during operation as well as the next, after stopping, disengagement of the discs and their oil cooling. Due to the negligible influence of the thermal sensitivity of 65H steel on temperature (Figure 5), the calculations were performed with the constant, adapted to the initial temperature, material properties. This stemmed from the relatively short heating time of the clutch, less than 7 s, and thus limited ability to heat conduction to other neighboring parts of the assembly. On the other hand, the cooling step following the friction heating and lasting ≈90  s took place in the environment of the oil, which absorbed heat from the surface of the components intensely compared to air. The construction stages of the second model are presented below.

### 6.1. Boundary Conditions

As mentioned above, the analyzed friction pair consisted of three geometric objects representing the basic elements of the clutch. The calculations were divided into two stages:Heating of the friction surfaces during sliding contact with convection cooling of the side free surfaces;Exclusive convection cooling of the lateral surfaces and working faces of the discs, where frictional heating occurred in the first stage. The second stage simulated the state when the components were disconnected (no friction).

It should be noted that in both stages the surfaces of the discs parallel to the friction surfaces were adiabatic.

In the first stage, during the frictional heating with duration time denoted $t_{s,i}$, i=0, 1, 2, 3, 4, the type of connection of geometric elements “create union” was used. This meant that the conditions of temperature continuity and heat flux intensity (perfect thermal contact) were required at the interface between the steel substrate and the clutch facing (friction material). On the other hand, on the contact surface of the friction material and the steel counterpart, there was a perfect thermal contact of friction, which consisted of meeting the following equality of:Temperatures of friction surfaces;The sum of the heat flux densities directed to each part and the boundary heat source power density.

On the lateral surfaces of both discs, heat exchange with the surrounding environment according to Newton’s law of cooling at the constant heat transfer coefficient $h=600\;\;W\;\;m^{-2}\;\;K^{-1}$ took place.

After stopping and disconnecting the clutch components, it was necessary to change the connection type of the parts in the “geometry” domain in COMSOL. Such a change affects almost all stages of the model creation (finite element mesh, selecting surfaces for heat transfer due to convection, etc.). Therefore, a new file was created, into which the temperature field from the last time step from the braking stage study was imported. Then, modifications were made to rebuild the geometry into an assembly. Creating an assembly, instead of the union formulation, allowed for the separation of the objects and the introduction of heat transfer due to convection also on the friction surfaces. The presence of such cooling better reflects the actual conditions in the clutch on the test bench. It was not possible when using the “create union” option in the computational model from the article [6].

### 6.2. Modeling Rotational Motion

As on the test stand, in the developed numerical models, it was assumed that the discs with the clutch facing are stationary, and the steel counterpart rotates at the angular velocity ω. The rotation of the counterpart in relation to the stationary disc was carried out using the well-known and verified approach of changing the velocity field at each point of the rotating part. The components of the linear velocity V vector were determined respectively from the dependence $V_{x}=-y\omega$ and $V_{y}=x\omega$ using a special tool available as the Translational Motion option of the Heat Transfer module of the COMSOL Multiphysics^®^ software (Heat Transfer in Solids-Solid-Translational Motion).

### 6.3. Construction of a Finite Element Mesh of the Clutch

Apart from the counterpart (steel disc) characterized by geometrical axial symmetry (mounting elements were omitted) (Figure 3b), there were differences in the shape in the circumferential direction of the other parts (steel plate with the clutch facing) (Figure 3a). The spatial (3D) model of the clutch was selected for the thermal finite element analysis.

When dividing the 3D geometric objects of the clutch into finite elements, an automatic mesh generator with an option of tetrahedral elements (free tetrahedral) and the general default size appearing under the name “normal” was used. This method takes into account the type of the problem as well as the curvature and geometric details that change mesh (divide into smaller elements) only in critical areas. Initial attempts to manually create mapped or free quad elements and then building regular hexagonal finite elements on the basis of the sweep method showed a number of warnings and errors at the edges of the objects. This was due to the large difference in the size of the contacting edges of the two parts—the smallest edges in the case of a friction material with many cuts on the working surface, steel plate, and the counterpart.

The final mesh created from of tetrahedral elements is shown in Figure 6, and before the actual calculations, it was additionally verified in terms of distributions and maximum values of temperature in the braking process.

### 6.4. Results of Computer Simulations

The second order shape function (quadratic Lagrange) of elements was used to calculate the temperature fields at both stages (heating and convection cooling). Such finite elements generated the most accurate results without the need to use an extremely fine mesh in the area of high temperature gradients. An experience in the construction of a finite element grid was obtained from previously conducted simulations of heat generation in disc brakes [25] and tread brakes (wheel-rail) of railway vehicles [26]. It was found that the linear finite elements significantly falsify the calculations (over 20%) even at many times higher than the default mesh density. The results of the calculations of the working surfaces temperature of the clutch are shown in Figure 7 and in Table 4.

In order to investigate the effect of oil cooling in the contact area, the temperature distributions of the clutch in the cross-section (r,z) were compared under the condition of perfect thermal contact and with the disconnected parts after stopping time moments $t_{s}+0\;\;s$, $t_{s}+15\;\;s$, and $t_{s}+55\;\;s$ (Figure 8, Figure 9 and Figure 10). It should be noted that because of different braking durations for each of the five friction materials, the presented distributions occur at slightly different points in time from the beginning at t=0  s.

The temperature distributions in Figure 8 show the stopping times $t_{s}+0\;\;s$. Slight differences in the distributions for variants a and b result from the fact that for connected clutch components (variant a) these are the values calculated and displayed from the model in which the perfect thermal contact condition was maintained all the time, while for disconnected components (variant b) the field is imported to the model with the separate cooling. The highest temperature is accumulated in the central part of the friction path near the friction radius.

Significant differences in temperature distributions resulting from the cooling method appeared after time $t_{s}+15\;\;s$ (Figure 9). Due to the smaller total cooled area of the clutch, at this time moment, significantly higher temperature values were achieved for the model in which the friction pair remained connected (Figure 9a). Only for materials 1 and 2, for which the shortest cooling times take place, was the maximum temperature of the friction disc for variant b equally high.

The temperature evolutions are confirmed by the temperature distributions shown in Figure 10. It can be clearly seen that the temperature field for each of the tested materials was similar. However, while at the time moment $t_{s}+15\;\;s$, a higher temperature was obtained for the friction disc, and at $t_{s}+55\;\;s$ a higher average temperature occurred for the steel disc.

## 7. Results and Discussion

The article presented an experimental analysis of material properties and thermal finite element analysis of friction heating for new PSFM materials used for clutch facing under lubricated conditions. Experimental tests were carried out on the IM-58 friction machine for four different PSFM materials with different additives (SiC, Ti-46Al-8Cr, and TiO_2_) and one base material. The materials produced were formed into friction discs and combined with a steel 65H disc, determining the values of the friction coefficients for each pair. The thermophysical properties for the new materials were investigated using the Hot Disk TPS2500S analyzer of thermal properties. These properties and values of the friction coefficient, as well as the input parameters (initial velocity, contact pressure, moment of inertia of rotating masses) of the experiment were adapted to 3D numerical models of friction heating. Based on the computer simulations carried out for the heating stage only with five friction materials, the temperature distributions (its maximum value on the contact surface and the time to reach this value), taking into account the temperature changes of the material properties of the steel disc, were analyzed. In the second part of the numerical tests, both the friction heating stage and the cooling stage after the clutch was disengaged were taken into account.

One of the main results of the material, tribological, and numerical tests carried out is the selection of such additives that had the greatest impact on the operation of the friction pair, and thus also on the temperature level during clutch engagement. It was shown that the greatest change of the tribological properties was obtained using addition of 3 wt.% SiC in the composition of the friction material based on copper with 12% tin and 30 vol.% graphite GE-1, namely, the coefficient of friction increased from 0.036 to 0.068. At the same time, wear increased from 3.1 to 13.5 $\mu m\;\;{\mathrm{km}}^{-1}$. The least influence on the tribological properties of the base material has 2 wt.% TiO_2_ powder, i.e. the coefficient of friction was the smallest (0.043) at the greatest wear resistance.

The basic factors influencing the changes in temperature distribution in the friction pair components and the evolution of the maximum temperature in the contact zone include (1) the amount of mechanical energy converted into heat, and thus the initial angular velocity and the moment of inertia of rotating masses; (2) the velocity at which this energy is dissipated, i.e. the braking torque dependent on the clamping force, coefficient of friction, and the friction radius; (3) type and dimensions of the given friction pair (thickness, number of neighboring elements absorbing heat), (4) thermophysical properties, and (5) cooling conditions due to convection and thermal radiation.

Assuming that in the analyzed friction pairs, braking takes place at the same input parameters (initial angular velocity, moment of inertia of rotating masses and clamping force), and assuming that the process time is short enough to ignore the influence of cooling, the key factors that affect the maximum temperature are the thermophysical properties and the coefficient of friction. As shown in Table 1, the thermophysical properties were very similar, while the greatest difference in the values of the friction coefficients was 89% (SiC in relation to the base material). Therefore, it is the coefficient of friction and the resulting braking time that in this case play a key role in reaching the maximum temperature value. For the higher coefficient of friction, the braking time is shorter, and the maximum temperature higher since the time for heat dissipation from the contact area due to conduction being limited.

The shortest braking time $t_{s,1}=3.61\;\;s$ and the highest temperature value equal to 160.2 °C among the five numerically tested materials was for the material with the addition of ceramic powder (SiC)—the greatest value of the coefficient of friction (Table 1). The longest braking time $t_{s,0}=6.82\;\;s$ and lowest temperature on the working surfaces, equal to 122.1 °C, was reached for the base material—the least value of the coefficient of friction.

Taking into account the disconnection of the clutch elements after stopping and convection cooling of the working surface with oil at the heat transfer coefficient *h* influences the value of the maximum contact surface temperature. The difference in the average temperature value for the five materials with the clutch components disconnected and the average temperature value obtained while maintaining the condition of perfect thermal contact was about 13 °C (30%) in the middle of the cooling stage (t≈40  s) and 5 °C (17%) at the end (t≈90  s).

Based on the presented research, we can conclude that the most promising from the point of view of achieving the shortest braking time with the same total friction work is the friction material with a 3% addition of SiC ceramics.

As a part of the future research, it is planned to determine the mechanical properties of the considered friction materials and to carry out numerical calculations of thermal stresses. In addition, attempts will be made to take into account the thermal contact resistance instead of using the perfect thermal contact condition.

## Figures and Tables

**Figure 1 materials-15-05384-f001:**
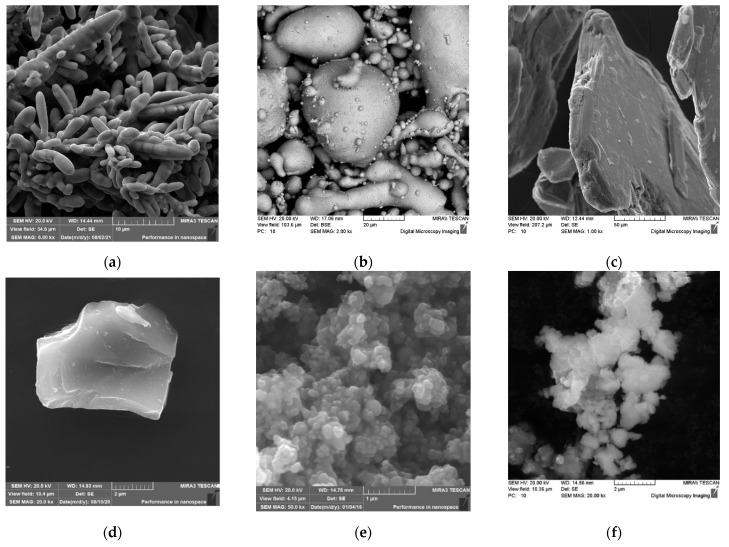
Morphology of the surface of the particles of powders: (**a**) the copper; (**b**) the tin; (**c**) the elemental graphite GE-1; (**d**) the silicon carbide; (**e**) the titanium dioxide; (**f**) the intermetalide Ti-46Al-8Cr.

**Figure 2 materials-15-05384-f002:**
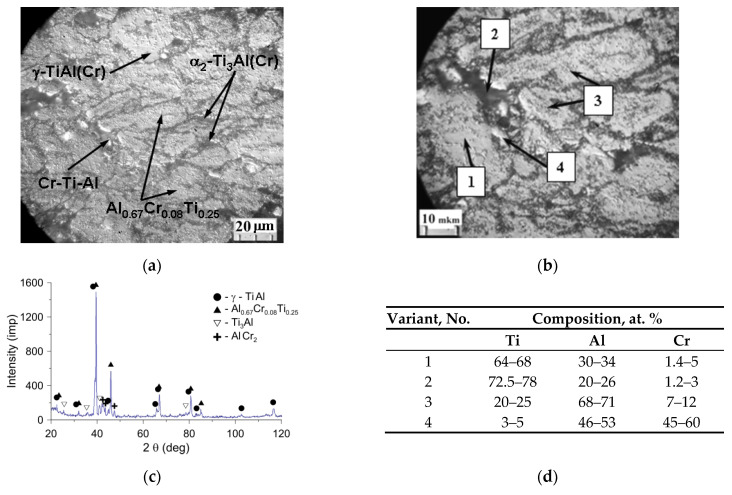
Microstructure and phase composition of the synthesized SHS powder of the Ti-46Al-8Cr system before grinding: (**a**,**b**) structure; (**c**) radiograph; (**d**) results of the micro X-ray spectral analysis (MRSA).

**Figure 3 materials-15-05384-f003:**
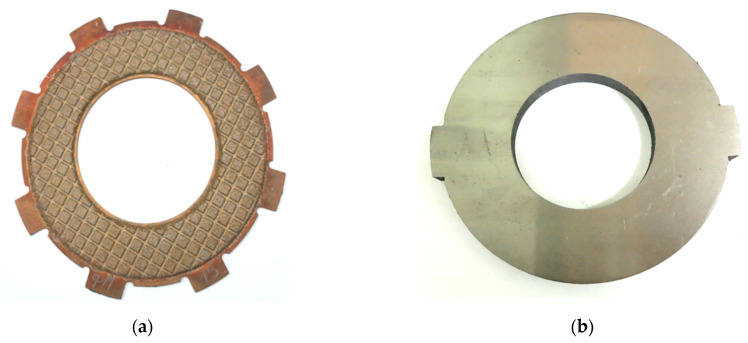
Clutch elements: (**a**) friction disc; (**b**) steel disc-counter body [6].

**Figure 4 materials-15-05384-f004:**
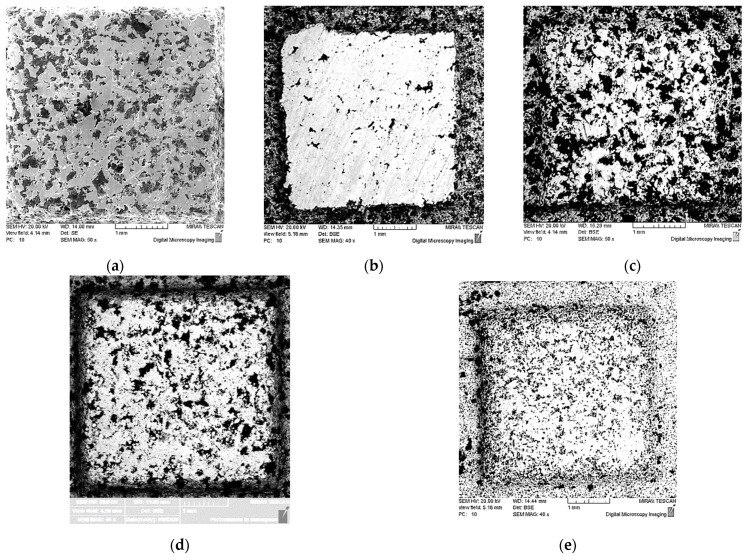
Morphology of the friction surface of the: (**a**) basic material; with powder additives (**b**) SiC; (**c**) Ti-46Al-8Cr; (**d**) 2 wt.% TiO_2_; (**e**) 5 wt.% TiO_2_.

**Figure 5 materials-15-05384-f005:**
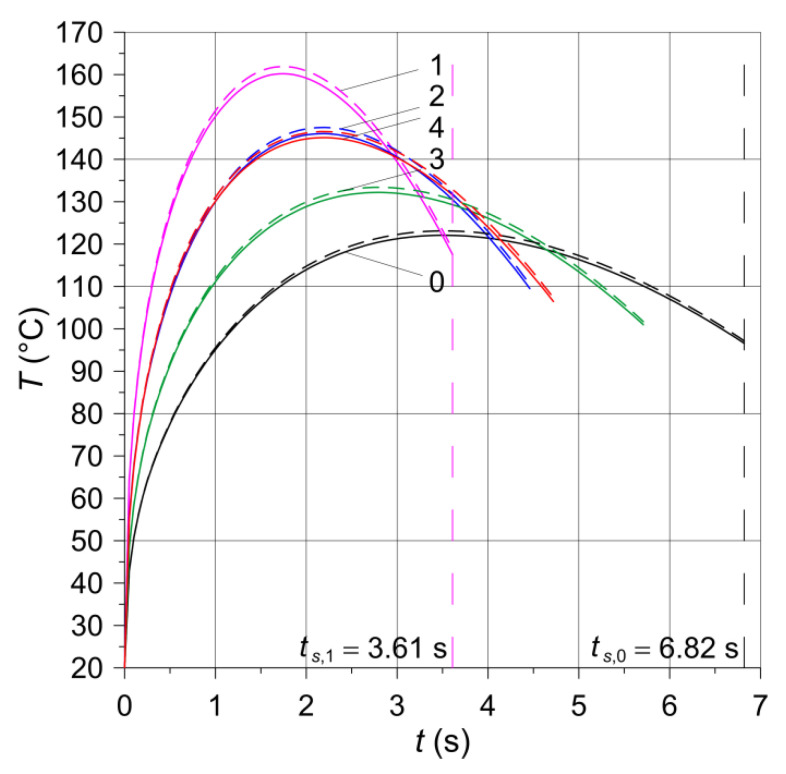
Evolution of the temperature of the friction surfaces of the clutch on the equivalent radius $r_{eq}=39.4\;\;\mathrm{mm}$ with constant (solid lines) and temperature-dependent (dashed lines) properties of 65H steel for five friction materials. Numbers 0–4 denote friction materials given in Table 1.

**Figure 6 materials-15-05384-f006:**
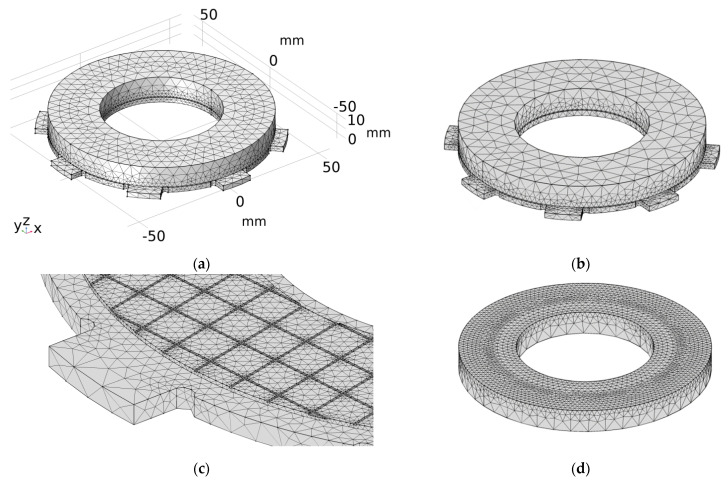
Finite element mesh used in finite element analysis of: (**a**) heating; (**b**–**d**) cooling.

**Figure 7 materials-15-05384-f007:**
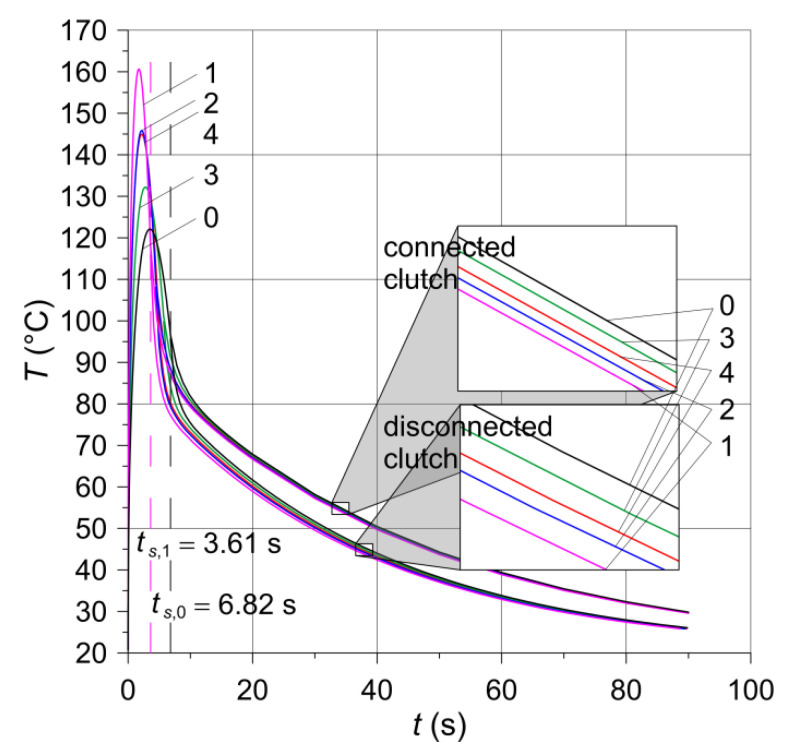
Evolutions of the temperature of the friction surfaces of the clutch on the equivalent radius $r_{eq}=39.4\;\;\mathrm{mm}$ obtained by means of the computational model: from the article [6]—connected clutch; developed in this paper—disconnected clutch. Numbers 0–4 denote friction materials given in Table 1.

**Figure 8 materials-15-05384-f008:**
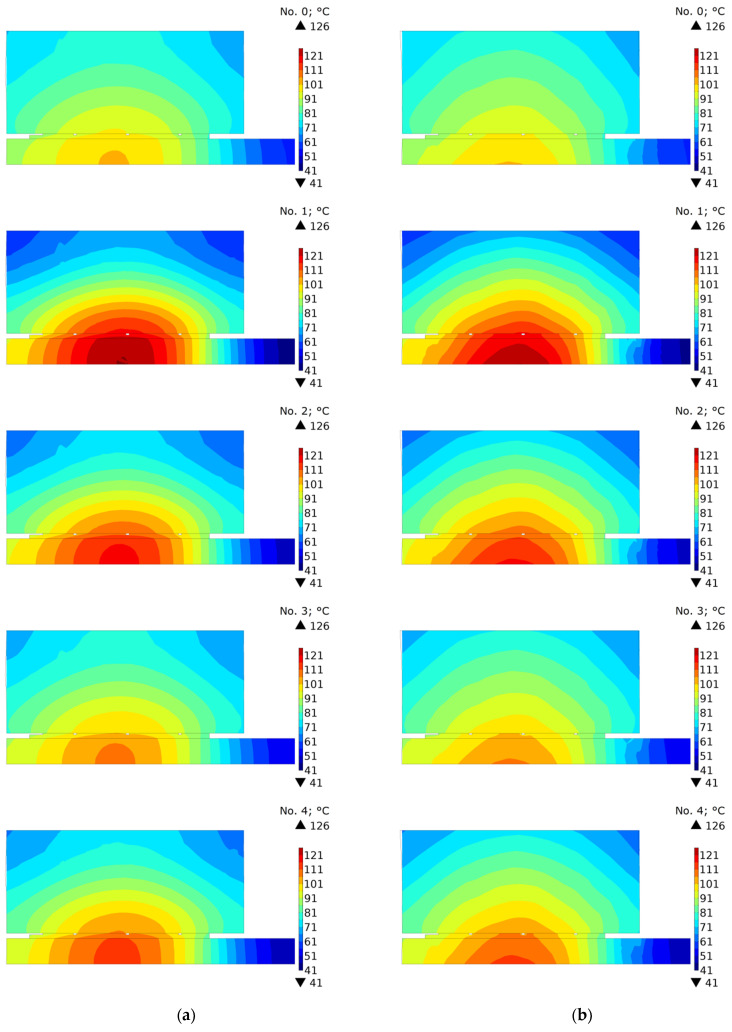
Temperature distribution at time $t_{s}+0\;\;s$ obtained using: (**a**) connected [6]; and (**b**) disconnected parts of the clutch for materials no. 0, 1, 2, 3, 4.

**Figure 9 materials-15-05384-f009:**
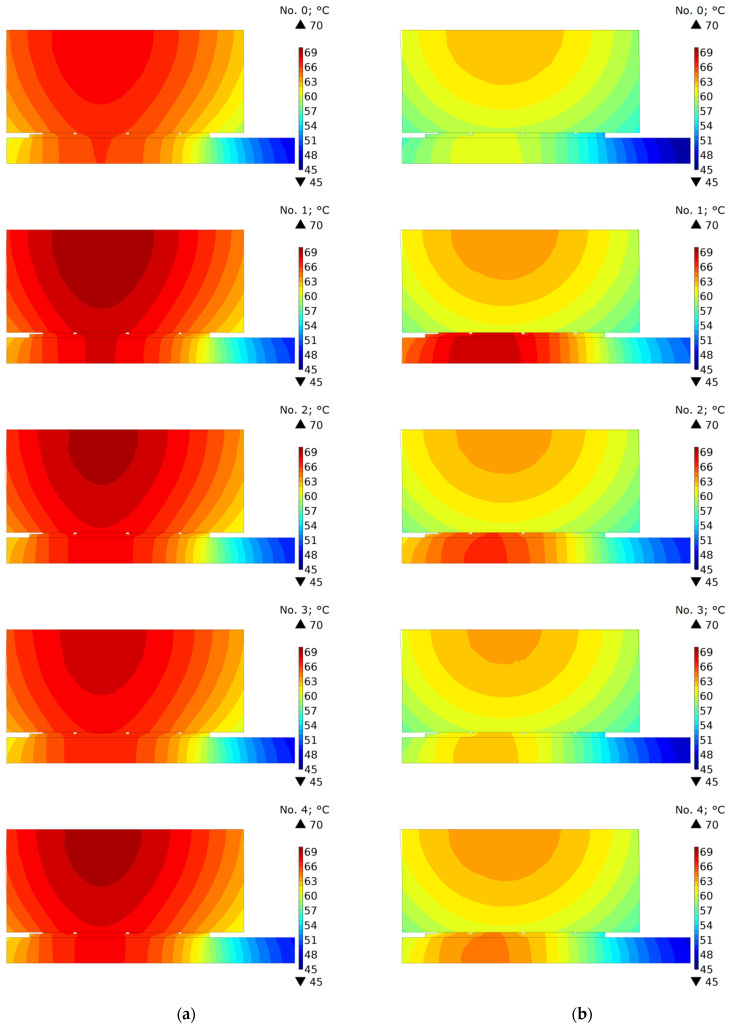
Temperature distribution at time $t_{s}+15\;\;s$ obtained using: (**a**) connected [6]; and (**b**) disconnected parts of the clutch for materials no. 0, 1, 2, 3, 4.

**Figure 10 materials-15-05384-f010:**
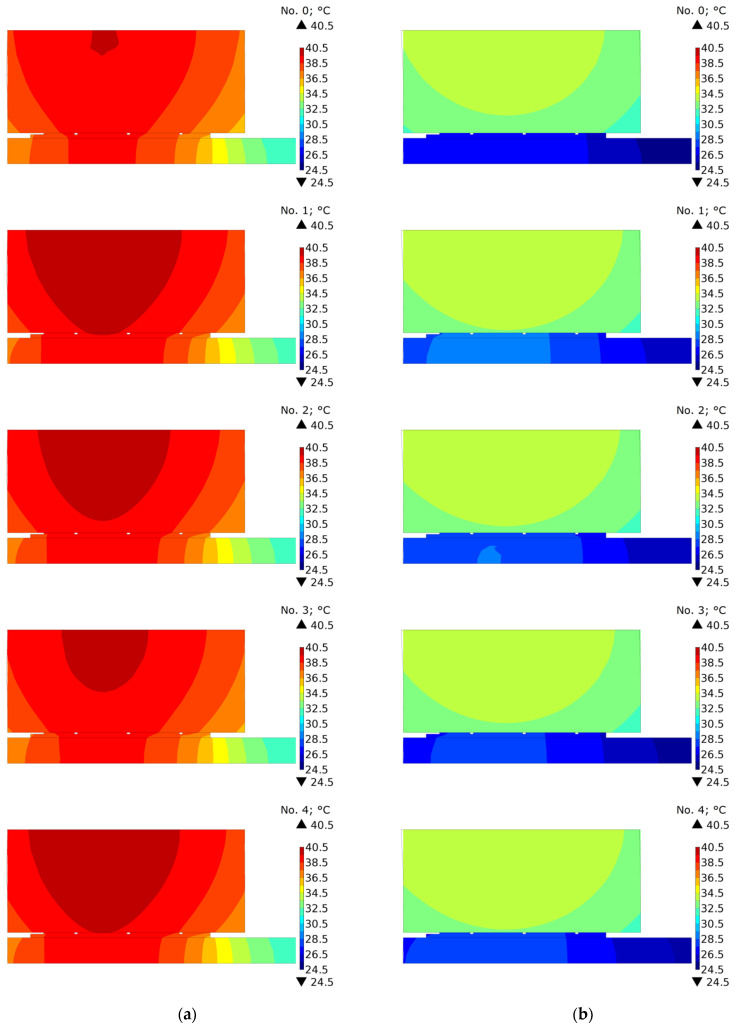
Temperature distribution at time $t_{s}+55\;\;s$ obtained using: (**a**) connected [6]; and (**b**) disconnected parts of the clutch for materials no. 0, 1, 2, 3, 4.

**Table 1 materials-15-05384-t001:** Influence of the type of carbon-containing additive on the thermophysical properties and coefficient of friction.

Variant, No.	Material	Thermal Conductivity, $W\;\;m^{-1}\;\;K^{-1}$	Specific Heat, $J\;\;{\mathrm{kg}}^{-1}\;\;K^{-1}$	Density, $\mathrm{kg}\;\;m^{-3}$	Coefficient of Friction, Dimensionless	$\mathrm{Wear},\;\mu m\;\;{\mathrm{km}}^{-1}$
0	Basic	7.9	5.4	4340	0.036	3.1
1	SiC 3%	6.5	5.4	4110	0.068	13.5
2	Ti-46Al-8Cr 2%	6.75	5.6	4220	0.055	5.1
3	TiO_2_ 2%	5.9	4.8	4210	0.043	3.6
4	TiO_2_ 5%	4.83	3.9	3980	0.052	4.3

**Table 2 materials-15-05384-t002:** Temperature-dependent properties of the steel 65H [24].

Temperature, °C	Thermal Conductivity, $W\;\;m^{-1}\;\;K^{-1}$	Density, $\mathrm{kg}\;\;m^{-3}$	Specific Heat, $J\;\;{\mathrm{kg}}^{-1}\;\;K^{-1}$
20	37	7850	490
100	36	7830	490
200	35	7800	510
300	34	7800	525
400	32	7730	560
500	31	7730	575
600	30	7730	590
700	29	7730	625
800	28	7730	705

**Table 3 materials-15-05384-t003:** Calculated parameters of the braking process for five friction materials.

Variant, No.	$t_{\max},\;s$	$T_{\max},\;{}^{\circ}C$	$t_{s},\;s$	$T_{s},\;{}^{\circ}C$
0	3.5	122.1	6.82	96.6
1	1.75	160.2	3.61	117.5
2	2.2	146.1	4.46	109.5
3	2.8	132.2	5.71	101.0
4	2.2	145.1	4.72	106.4

**Table 4 materials-15-05384-t004:** Calculated parameters for the cooling stage for five friction materials.

Variant, No.	$t_{s},\;s$	$t_{c},\;s$	$T_{c}^{\left(c\right)},\;{}^{\circ}C\;\mathrm{Connected}\;$ [6]	$T_{c}^{\left(d\right)},\;{}^{\circ}C\;\mathrm{Disconnected}$
0	6.82	83.18	29.8	22.2
1	3.61	86.39	29.6	25.6
2	4.46	85.54	29.6	25.7
3	5.71	84.29	29.8	25.9
4	4.72	85.28	29.7	25.8

## Data Availability

Not applicable.

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
