# Peer review of "Influence of the Additive of Ceramic and Intermetallic Powders on the Friction Properties and Temperature of the Wet Clutch Disc"

_materials, 2022, doi:10.3390/ma15155384_

Round 1

Reviewer 1 Report

After carefully reading the article, the reviewer must make a series of essential comments to improve the quality of the manuscript. The comments are:

-        - The abstract should be modified. Firstly, there should be an introduction to the problem and the novelty of the research, then the methodology followed should be detailed, followed by the results obtained and, finally, the conclusions drawn from the research.

-        -  In the Introduction section, the novelty of this research and the fundamental basis on which it is based should be clearly detailed. In other words, detail the problem that this research attempts to solve.

-       -   In the section on materials and methods, the initial materials used should be defined in much more detail.

-       -   In the materials and methods section, the methods used in the different tests and in the modelling should be defined in much greater detail.

-      -    The reviewer considers that there should be a section entitled Results and discussion that brings together sections 3, 4, 5 and 6. The relationship between the different sections should also be clearly shown.

-      -    Conclusions are too superfluous, so it must be defined with quality.

If the authors make the revisions discussed above, the article may be reconsidered for publication, however, the current format is not clear for publication.

Author Response

We would like to thank the referees for their comments, and we are pleased to improve the manuscript according to their suggestions. We have provided a point-by-point response below to feedback and have modified the text (shown in the manuscript in BLUE).

Comment 1: The abstract should be modified. Firstly, there should be an introduction to the problem and the novelty of the research, then the methodology followed should be detailed, followed by the results obtained and, finally, the conclusions drawn from the research.

Answer 1: The comment has been taken into account in the new version of the manuscript.

Comment 2: In the Introduction section, the novelty of this research and the fundamental basis on which it is based should be clearly detailed. In other words, detail the problem that this research attempts to solve.

Answer 2: The comment has been taken into account in the new version of the manuscript.

Comment 3: In the section on materials and methods, the initial materials used should be defined in much more detail.

Answer 3: The comment has been taken into account in the new version of the manuscript.

Comment 4: In the materials and methods section, the methods used in the different tests and in the modelling should be defined in much greater detail.

Answer 4: The comment has been taken into account in the new version of the manuscript. We added more information about the carried out research.

Comment 5: The reviewer considers that there should be a section entitled Results and discussion that brings together sections 3, 4, 5 and 6. The relationship between the different sections should also be clearly shown.

Answer 5: The comment has been taken into account in the new version of the manuscript.

Comment 6: Conclusions are too superfluous, so it must be defined with quality.

Answer 6: We have decided to move the conclusions into new combined section Results and discussion.

Reviewer 2 Report

Reviews

In the review of research article titled:” Influence of the additive of ceramic and intermetallic powders on the friction properties and temperature of the wet clutch disc”, article is designed very well and explanation is also good. I would like to see this article publish but after some minor modifications;  

1- Along with MSRA test, Please provide some EDS or any other compositional analysis test to accurately/precisely measure the composition of different metals and ceramics in the studied samples.

2- Provide a table comparing the friction coefficient of metal oxide TiO2 with previous results so that improved values can be compared easily.

3- Why the highest temperature value equal to 160.2 ° C, was obtained for the material with the addition of SiC powder only. Elaborate it with some references in results part.

4- What are the key parameters effecting the contact surface temperature must be studied and presented thoroughly.

5- Can you please present a numerical calculation model for the thermal stress too so that it can be easily understandable.

Author Response

We would like to thank the referees for their comments, and we are pleased to improve the manuscript according to their suggestions. We have provided a point-by-point response below to feedback and have modified the text (shown in the manuscript in BLUE).

Comment 1: Along with MSRA test, Please provide some EDS or any other compositional analysis test to accurately/precisely measure the composition of different metals and ceramics in the studied samples.

Answer 1: Similar data on the chemical composition of the intermetalide Ti-46Al-8Cr are given in the work https://doi.org/10.29235/1561-8323-2021-65-1-103-110. Additional data are not available at that moment, this requires repeating the studies.

Comment 2: Provide a table comparing the friction coefficient of metal oxide TiO2 with previous results so that improved values can be compared easily.

Answer 2: It has been experimentally established that with a TiO2 content of more than 5%, the wear of the friction material increases. In our opinion, this is due to the decrease in the strength of the intergrain boundary of the bronze matrix. With a TiO2 content of less than 2%, there are no changes in tribological properties, i.e. the values of the coefficients of friction and wear do not change. Thus, 2% and 5% are accepted as a boundary, and are given in the results of this article.

Comment 3: Why the highest temperature value equal to 160.2 °C, was obtained for the material with the addition of SiC powder only. Elaborate it with some references in results part.

Answer 3: The basic factors influencing the changes in temperature distribution in the friction pair components and the evolution of the maximum temperature in the contact zone include 1) the amount of mechanical energy converted into heat, and thus the initial angular velocity and the moment of inertia of rotating masses; 2) the velocity at which this energy is dissipated, i.e. the braking torque dependent on the clamping force, coefficient of friction and the friction radius; 3) type and dimensions of the given friction pair (thickness, number of neighboring elements absorbing heat), 4) thermophysical properties, and 5) cooling conditions due to convection and thermal radiation.

Assuming that in the analyzed friction pairs, braking takes place at the same input parameters (initial angular velocity, moment of inertia of rotating masses and clamping force), and assuming that the process time is short enough to ignore the influence of cooling, the key factors that affects the maximum temperature are the thermophysical properties and the coefficient of friction. As shown in Table 1, the thermophysical properties are very similar, while the greatest difference in the values of the friction coefficients is 89% (SiC in relation to the base material). Therefore, it is the coefficient of friction and the resulting braking time that in this case play a key role in reaching the maximum temperature value. For the higher coefficient of friction, the braking time is shorter, and the maximum temperature higher since the time for heat dissipation from the contact area due to conduction is limited.

The answer given above has been enclosed in “Results and discussion” section.

Comment 4: What are the key parameters effecting the contact surface temperature must be studied and presented thoroughly.

The answer to this question is contained in the reply to comment 3.

Comment 5: Can you please present a numerical calculation model for the thermal stress too so that it can be easily understandable.

Answer 5: At the end of the article, we included a sentence regarding our future research plans related to the development of a numerical model for the determination of thermal stresses in friction clutch elements. This article does not address this issue.

In the currently being developed coupling model, the quasi-static contact problem is being solved. This means that the model takes into account the interdependence of the non-uniform pressure distribution, thermal expansion of the materials of the contacting components and the temperature in the contact zone at a specific change in angular velocity and a given coefficient of friction obtained from the experimental studies.